# Location, Location, Location: Design Bias with Recursive Kernel Transformation

## Abstract

It has been hypothesized that the old brain was compressed into cortical columns of the neocortex during the evolution of mammalian brains. Computational modeling of hippocampal-cortical interaction inspires us to propose a navigation-based implicit representation for manifold learning. The key new insight is to transform any explicit function (or geometrically a manifold) to an implicit representation using design bias for exploiting the concentration of measure (CoM) in high dimensional spaces. CoM-based blessing of dimensionality enables us to solve the manifold learning problem by direct-fit or local computation with guaranteed generalization property and without the need to discover global topology. We construct a memory encoding model, namely specification-before-generalization (SbG), and extend it into recursive kernel transformation to mirror the nested structure of the physical world. The biological plausibility of SbG learning is supported by its consistency with the wake-sleep cycles of mammalian brains. Finally, we showcase the application of design bias and recursive kernel transformation to understanding the phylogenetic continuity of navigation and memory and the manifold untangling of object recognition by the ventral stream.

## 1 Introduction

When Alan Turing first conceived the problem of artificial intelligence (AI), he took a top-down approach and proposed an imitation game known as the Turing test Turing (2009). Toward passing the Turing test, optimization-based machine learning, as exemplified by the training of various neural networks (including the latest transformer architectures Vaswani et al. (2017)), has dominated over the past decade. Latest advances in the field of AI, such as foundation models Bommasani et al. (2021) and AgentAI Durante et al. (2024), have further stimulated the interest in pushing for larger models (e.g., the evolution of OpenAI's ChatGPT models) trained on more data (e.g., the combination of vision with language data to support multimodal interaction). Little doubt was cast regarding their fundamental limitations, such as the notorious bias-variance dilemma Geman et al. (1992); Friedman (1997), its related curse of dimensionality Bellman (1966), or the sustainability of existing AI research Van Wynsberghe (2021).

It is time to pause and consider the alternative. The human brain can do marvelous things with 20 watts of power. A bottom-up approach to understanding the mechanism of the brain - e.g., sensorimotor knowledge learning in the emerging neuroAI paradigm Zador et al. (2022) might offer a fresh new perspective on AI or machine learning. As insightfully advocated in Geman et al. (1992), "the fundamental challenges in neural modeling are about representation rather than learning per se." Vernon Mouncastle's discovery about column organization Mountcastle (1957) in the 1950s inspired the pursuit of a universal cortical processing algorithm. It has been hypothesized in Hawkins (2021) that "nature stripped down the hippocampus and entorhinal cortex to a minimal form (cortical columns)". The more rapid expansion of the neocortex than the hippocampus during the evolution of mammalian brains is the key to understanding the adaptive behavior of mammals (the origin of natural intelligence).

The above hypothesis suggests the possibility of constructing a universal learning representation based on the modeling of hippocampal-neocortical interaction. It is long known that the neocortex can detect or predict orderly patterns in the external environment, while the hippocampus plays the role of the neocortex's librarian Buzsáki (2006)). Such observation inspired us to draw an anal-

ogy between cortical columns and the simultaneous localization and mapping (SLAM) model Dissanayake et al. (2001), as shown in Fig. 1. Using estimation theoretical solution to the SLAM problem as the building block, one can rigorously show that cortical columns, when connected with sensory and motor systems, become intelligent agents capable of learning local maps by establishing the association between where (locations) and what (landmarks).

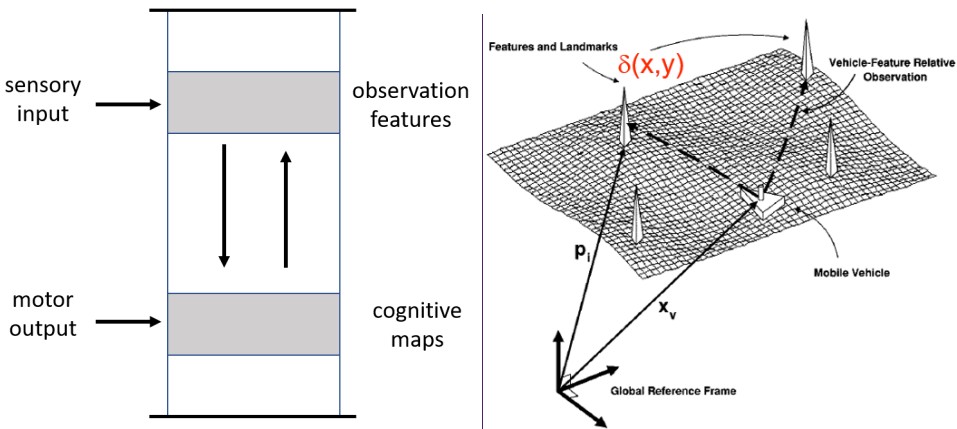

Figure 1: Biological inspiration behind this work: we draw an analogy between cortical columns (credit: Hawkins (2021)) and the SLAM model (adapted from Dissanayake et al. (2001)).

Based on the above analogy, an important new insight brought by our approach is that deictic codes Ballard et al. (1997), a kind of pointer mechanism indicating the location on cognitive maps Whittington et al. (2022), can serve as design bias Geman et al. (1992) to facilitate the exploitation of concentration of measure (CoM) in the latent space. As the CoM theory Talagrand (1995) demonstrates, the $\epsilon$-ball expansion of any event with a probability $\geq \frac{1}{2}$ can cover almost the entire space. Such observation implies the feasibility of solving the manifold learning problem locally with good generalization property (turning extrapolation into interpolation in the high-dimensional space) and without the knowledge of global topology (a.k.a. intelligence without representation Brooks (1991) or direct-fit Hasson et al. (2020)). Such a strategy of CoM-based manifold learning can be implemented by introducing spatio-temporal context variables known as *design bias* Geman et al. (1992) that mimic the indexing by the hippocampus-entorhinal cortex system. The idea of specification (i.e., through context induction) before generalization (e.g., context aliasing McCallum (1995)) reflects a computational abstraction of the two-stage memory formation and consolidation proposed by Buzsaki Buzsáki & Moser (2013). From a manifold learning perspective, specification with design bias linearizes the manifold locally by change of coordinates such that the local geometry in the latent space (as indexed by the contextual variables) can be discovered by the SLAM agent. The critical question is what kind of representation can be learned by the SLAM agent as a computational abstraction of cortical columns.

In this paper, we advocate for the class of implicit representations such as delta measures Sanders & Yokoyama (2012) as a universal solution to CoM-based biased learning. We first show how to implicitly represent any function via parameterization so any object $X$ in the sensory space can be transformed into a probabilistic measure in the latent space. To reach above chance ($P(A) \geq \frac{1}{2}$) as required by CoM, we introduce specification-before-generalization (SbG) as a biologically plausible memory encoding and consolidation strategy based on the simple fact $P(X|C) > P(X,C)$ (i.e., conditioning boosts the CoM). Unlike Markovian processes (e.g., Markov Chain), we consider a new model that recursively encodes contextual/conditioning variables (design bias) into episodic memory by specification; the corresponding generalization operator can be implemented by integrating contextual variables out (a computational abstraction of semantic memory) Buzsáki & Moser (2013). Under the framework of CoM-based implicit representation, recursive encoding attempts to mirror the hierarchical organization of the external environment by multi-scale contextual variables, which extends the existing deictic codes Ballard et al. (1997) and generalized Hough transform (GHT) Ballard (1981). To justify the biological plausibility, we briefly discuss the connections of this work with sensory (visual and haptic) perception during the evolution and development of mammalian brains. In summary, we make the following contributions in this paper:

- **Navigation-memory analogy**. Based on the hypothesis of phylogenetic continuity of navigation and memory, we show how the evolution of mammalian brains can be better understood from the perspective of navigation in the latent space. Cortical columns are computationally abstracted by SLAM agents for local map making;

- **Specification-before-Generalization (SbG) learning**. Using spatiotemporal context as design bias, SbG learning is conceptually simple (conditioning boosts CoM) and biologically consistent with the sleep-wake cycle in nature. Moreover, we rigorously show that any nonlinear manifold can be decomposed into the superposition of local maps by conditioning for exploiting CoM or blessing of dimensionality.

- **Implicit representation via kernel transformation**. Using delta measures as the pivot for cortical transforms, we show how to obtain implicit representation for an arbitrary content or context variable by kernel transformation. A side benefit of kernel transformation is that it simultaneously encodes content and context information as peer variables in the latent space to support task-dependent representation learning.

- **Hierarchical representation via non-Markovian memory encoding**. We show how to hierarchically extend kernel transformation to mirror the external environment's nested structure by non-Markovian memory encoding. Such map-in-map hierarchy allows an organism to develop adaptive behavior by local computation (direct-fit) only without the need of discovering global topology.

## 2 TOY PROBLEM AND GEOMETRIC INTUITION

The objective of a general learning problem is to predict the response vector $\boldsymbol{y}$ from the feature/input vector $\boldsymbol{x}$. Optimization-based approaches to learning attempt to construct a machine/function $f$ such that some pre-selected cost functional is minimized Geman et al. (1992). However, as shown in Geman et al. (1992), those approaches suffer from the fundamental tradeoff between bias and variance, which describes the relationship between a model's complexity, the accuracy of its predictions, and how well it can make predictions on previously unseen data that were not used to train the model. It has been argued in Gigerenzer & Brighton (2009) from a cognitive science perspective that the human brain resolves this notorious dilemma in the case of the typically sparse, poorly characterized training sets by adopting high-bias/low-variance heuristics (i.e., "less-is-more" as the justification of heuristic-based learning). More rigorously, one can argue the only way to overcome this barrier is to "prewire important generalizations" or "purposefully introduce bias" Geman et al. (1992). Such design bias could allow an organism to discover the nature of the biases "internalized during the course of evolution. Geometrically, our intuition is to facilitate the task of classification (i.e., "understand the world as it is") but not regression (i.e., "understand the world as it appears") by high-bias/low-variance heuristics. To formalize such geometric intuition, we start from the following twisted puzzle in computational geometry Minsky & Papert (2017).

**Computational Geometry Puzzle**: Given $\boldsymbol{x}_i = (x_1^i, x_2^i) \in R^2$ and class label $y_i \in \mathcal{N}$ ($i = 1, 2, ..., N$), does there exist a universal and optimal solution to transform folded data manifolds as shown in Fig. 2 into linearly separable Dirac's delta functions in the latent space?

A moment of thought shows that the answer is Yes but nontrivial. For example, the kernel trick or neural network will not work here because there is no generic rule for kernel construction or neural network that works for data manifolds with arbitrary topology and varying folding strategies (note that a different way of paper folding will change the definition of geodesic distance in the folded space). The new attack is to treat class label $y_i$ as a *contextual* variable (induced bias) and solicit a linear dichotomy in the lifted 4D space $(x_1^i, x_2^i, x_3^i, y_i) \in R^4$. In the case of binary classification, the hyperplane $y = \frac{1}{2}$ can separate two classes regardless of the global topology of data manifolds or the way of folding into the paper ball. Note that such a lifting-based idea essentially exploits the blessing of dimensionality by folding the data distribution along an unoccupied dimension.

**Context as Design Bias**: One can argue that the above lifting-based specification is useless if it does not generalize. Specification and generalization are two sides of the same coin for memory Brunel et al. (2009) and learning D'Amour et al. (2022). Design bias can be implemented by conditioning based on a simple observation $P(\boldsymbol{x}|y) > P(\boldsymbol{x}, y)$ since $P(y) < 1$. Taking the three-circle manifold as an example but with another twist of physics-based conditioning - *paper folding mimics the embedding into a higher dimensional space where distortion is inevitable*. That is, we assume

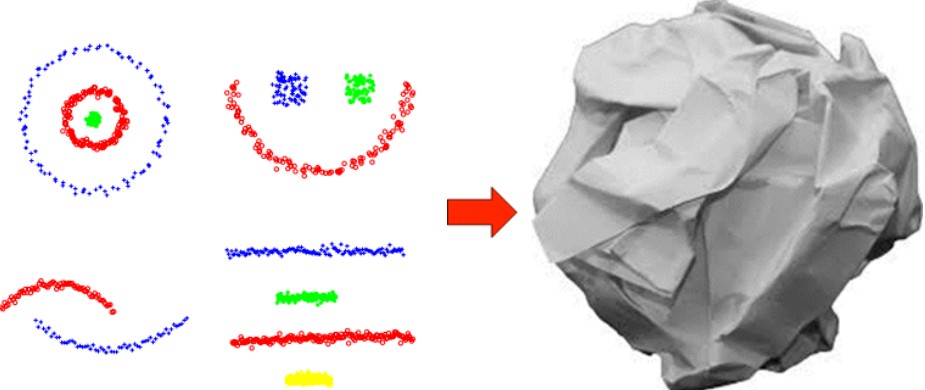

Figure 2: Toy examples of data manifolds in 2D - after folding into 3D (Image credit: adapted from the Internet), does there exist a universal solution to represent and separate different classes (data points labeled by different colors) regardless of the folding strategy? In this paper, we provide affirmative answers and show its connections with universal learning via implicit representations.

$(x_1^i, x_2^i)$ are recorded positions of a mouse running in three circular mazes with varying radii (marked by red, green, and blue in the first example in Fig. 2). However, we can only observe the distorted trajectory data (i.e., folded into $R^3$) from which low-dimensional manifold constraints cannot be explicitly enforced. In this thought experiment, the only logical solution to the puzzle is to first faithfully track the trajectory of data points during the paper folding as a kind of episodic memory and then generalize the discovery by restoring the manifold constraint.

The above specification-before-generalization (SbG) solution can be connected with the strategies of direct fit Hasson et al. (2020) through implicit representations Sanders & Yokoyama (2012). Direct fit reflects a principle of local computation without assuming a global fitting function for optimization. Instead, it uses overparameterized models to decompose a manifold into local maps or locally piecewise linear regions (i.e., folded trajectory in the toy example). One salient feature of direct fit is that it allows interpolation instead of extrapolation in high-dimensional space (useful for generalization). An important new insight this work brings is *to facilitate the task of direct fit by design bias* based on the analogy between cortical columns and SLAM agents. By storing the indexes of local maps (isomorphic to Euclidean space and suitable for SLAM-based learning) in a global atlas, one can "navigate" on the manifold using local maps only without knowing the global topology (such view is consistent with "intelligence without representation" Brooks (1991) that uses the world as its own model). Conceptually similar to GPS-based navigation in the real world, direct-fit counts on local computation to interpolate among samples for generalization purposes. Specification or overfitting introduces a harmless bias for local computation by "remembering" the position/indexing of local maps within the global atlas (analogous to the interaction between the hippocampus and the neocortex). The result of SbG is the decomposition of any nonlinear manifold into linearly separable regions to facilitate representation (instead of optimization) in the latent space.

**Implicit representation**: SbG with design bias does not solve the problem of representation learning but facilitates the promotion of CoM by conditioning (contextual variables). Following the less-is-more principle Gigerenzer & Brighton (2009), we conjecture that the take-the-best heuristics (a.k.a. winner-takes-all Riesenhuber & Poggio (1999)) requires an implicit representation of content such as delta measures as the computational abstraction of polychronization neural groups Izhikevich (2006). From matched filters Wehner (1987) to decision-making Gigerenzer & Brighton (2009), both sensory and motor systems in cognitive systems must adapt to dynamically varying environments based on the principle of ecological rationality. Despite the structural diversity of the environment, it will be intellectually appealing if the external environment (both context and content) can be encoded into a universal representation, as advocated by Mountcastle's uniformity principle of cortical columns.

The idea of encoding an arbitrary object into Dirac's delta function dates back to Hough transform Duda & Hart (1972). Later, generalized Hough transform (GHT) Ballard (1981) was developed for detecting arbitrary shapes. An important new insight brought by our approach is that matched filters

(MF) can implement the voting mechanism adopted by GHT Wehner (1987) or, more generally, as a tool of CoM in the latent space. When combined with design bias, kernel-based MF can be interpreted as an implicit representation of episodic memory for encoding sensory signals. Delta measures naturally become the mathematical abstraction for the ideal response of kernel-based MFs, which pushes the probability measure of a response function locally concentrated in the latent space to exploit the benefit of CoM phenomenon. After reaching above the threshold ($\frac{1}{2}$), generalization can be guaranteed by interpolation (instead of extrapolation) in a high-dimensional space thanks to CoM.

## 3 LESS-IS-MORE: DESIGN BIAS FOR VARIANCE REDUCTION

In this section, we develop a theoretical foundation for less-is-more heuristics Gigerenzer & Brighton (2009). Design bias can improve the accuracy of classification by variance reduction, which is conceptually connected with the phenomenon of concentration of measure (CoM) Ledoux (2001). Based on the ecological rationality of heuristics, an organism can adapt to the external environment by purposefully introducing bias as prewired generalization Geman et al. (1992). If a greater reduction in variance can offset design bias, such heuristics can result in better inference with less processing (i.e., less-is-more heuristics). In this section, we first discuss how to design bias via specification-before-generalization (SbG) and study various variance reduction strategies for SbG. Then, we present how design bias is consistent with the less-is-more heuristics and CoM theory. Based on the nearest neighbor classifier Cover & Hart (1967), we will show how to achieve optimal classification asymptotically without the barrier of bias-variance tradeoff.

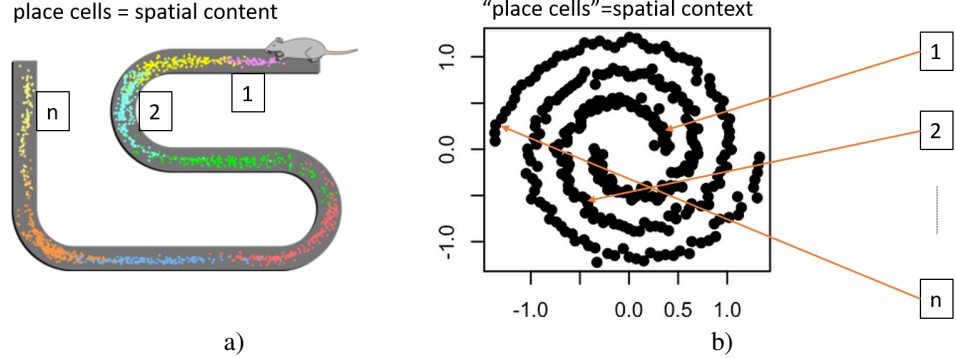

Figure 3: Design bias via contextual dependency. a) The place cells O'keefe & Nadel (1978) provide spatial content for navigation (credit: Wikipedia). b) The purposeful introduction of spatial context via "place cells" can decompose a nonlinear manifold into locally linear segments with CoM.

**Design bias via context dependency.** Following the toy example, we note that design bias can be induced by the spatiotemporal context, which varies from task to task. For example, place cells serve as spatial content for navigation (temporal context is defined by the trajectory passing through each position), as shown in Fig. 3a. However, the position/location can become the spatial context when we decompose a nonlinear manifold into the superposition of locally linear segments (refer to Fig. 3b). We do not need to discover the global topology for a classification (instead of regression) task because it is irrelevant. By analogy, we store an atlas of $n$ local maps, each of which records the biased information about spatial context $\mathcal{N}_i$ (note that $n$ can be even bigger than $N$ implying overparameterization) but resort to some indexing/pointing mechanism (e.g., associative memory Hopfield (1982) and sparse distributed memory Kanerva (1988)) for efficient retrieval. In summary, design bias represents a strategy of specification-before-generalization (SbG) for linearizing the learning problem. A direct consequence of linearization is the reduction of variance, which alleviates the tradeoff between bias and variance.

**Variance reduction strategies.** Generally speaking, there are three classes of strategies for variance reduction with design bias: 1) spatial averaging is based on the law of large numbers (i.e., $Var(\frac{1}{N}\sum_i X_i) = \frac{1}{N}Var(X)$); 2) temporal context helps resolve uncertainty (i.e., $P(\boldsymbol{x}|y) =$

$\frac{P(\boldsymbol{x},y)}{P(y)} > P(\boldsymbol{x},y)$); 3) nearest neighbor classifier Cover & Hart (1967) in space or time is known to contain at least half of the classification information. Exploiting these observations recursively can effectively reduce variance at an exponential rate.

Instead of learning $\phi : \boldsymbol{x} \to y$, we formulate supervision signal $y$ as a temporal context variable (a peer of spatial context variable $c$) and translate clustering into the problem of navigating on the data manifold. Without the knowledge about the global topology, we decompose the data manifold into a finite collection of local maps/neighborhoods (i.e., mixture of Gaussians) $\mathcal{N}_i = \{\boldsymbol{x}_k : \|\boldsymbol{x}_k - \boldsymbol{x}_i\| < \epsilon\}, i = 1, 2, ..., N$, which is similar to the finding of kNN for $\boldsymbol{x}_i$ in the first step of locally linear embedding (LLE) Roweis & Saul (2000). For each neighborhood, we store the $k + 1$ data points by an associative memory: $(\boldsymbol{x}_j, y_j|c_j)$ where $j \in \mathcal{N}_i$, which is called a *map*. The collection of $N$ maps forms an atlas corresponding to the knowledge about the global topology. Note that any inference task only requires local knowledge from a specific map instead of the global knowledge of the entire atlas. The complete algorithm is summarized into **Algorithm 1** below.

---

**Algorithm 1:** SbG-based Manifold Learning with an Atlas of Maps

---

**Input:** $\mathbf{X} = \{\boldsymbol{x}_1, ..., \boldsymbol{x}_N\}$ and $\mathbf{Y} = \{y_1, ..., y_N\}$

1 **Learning**: for each $\boldsymbol{x}_i$, find its kNN to generate $\mathcal{N}_i$ and associate it with a unique spatiotemporal context variable $c_i$ (map index);

2     -Locally, store $N$ maps $(\boldsymbol{x}_j, y_j)|c_i$ where $j \in \mathcal{N}_i$ is the kNN of $\boldsymbol{x}_i$ for $i = 1, ..., N$ using associative memory such as Universal Hopfield Network Millidge et al. (2022);

3     -Globally, store $N$ indexing $(\boldsymbol{x}_i|c_i)$ into another associative memory (the global atlas containing all local maps);

4 **Inference**: For an inquiry $\boldsymbol{x}_{new}$, first retrieve which map $c_i$ in the atlas by associative recall and save this map's index $I = c_i$;

5     - Go to local map with the index of $I$ to retrieve the corresponding class label $y_{I,J}$;

6     - Output the classification result $y_{new} = y_{I,J}$;

---

Given a new data point $\boldsymbol{x}_{new}$, we first determine which map(s) to use by global associative recall and then use this index $I$ to retrieve the corresponding class label $J$ from the local map. Such local-global coordination mimics the interaction between mammalian brains' neocortex and hippocampus Buzsáki (2006). To generalize SbG-based learning, we can collapse each local neighborhood $\mathcal{N}_i$ by nonlinear dimensionality reduction, which generalizes the known aliasing operator in cognitive maps George et al. (2021); Whittington et al. (2022). The basic idea behind the aliasing operator is to transform context-dependent representation $P(\boldsymbol{x}, y|c)$ into context-independent representation $P(\boldsymbol{x}, y)$. Algorithm 1 can be easily extended to support bagging or boosting by adopting $k$-NN classification when retrieving the local maps from the global atlas. From $k$ best-matching maps, we can first generate $k$ classification results and then take a majority vote, which can further improve the generalization performance. Formally, we have the following result.

**Proposition 1. (Asymptotically Optimal Classification of SbG-Based Learning**.

*Under the assumption with sufficient memory capacity for storing local maps and global atlas, the performance of Algorithm 1 can asymptotically achieve Bayes probability of error as $N \to \infty$.*

*Sketch of the proof.* It has been shown in Cover & Hart (1967) that twice the Bayes probability of error bounds the error made by NN classifiers. Under ideal circumstances, NN classifiers can achieve the Bayes probability of error as the size of the training dataset approaches infinity. Similarly, a linear classifier can achieve the Bayes probability of error when the classes are Gaussian with equal covariance matrices. Asymptotically, we can double the sample size recursively (parameter $N$ in Algorithm 1). Selectively sampling the manifold can gradually approximate each local map with a Gaussian distribution, and the data become linearly separable after specification. As $N \to \infty$, we observe that Algorithm 1 converges to both NN classifier and linear classifier with an identical minimum probability of error.

*CoM-based interpretation.* Algorithm 1 eliminates the bias-variance tradeoff by implicitly sampling the local neighborhood of important regions and recording their position within the global coordinates as the design bias. Thanks to the phenomenon of CoM Talagrand (1995), any $\epsilon$-ball expansion of a probabilistic event $A$ with $P(A) \geq \frac{1}{2}$ will cover almost the entire space. Therefore, even an approximate NN classifier will be sufficient for the classification task with guaranteed approximation

quality (i.e., large deviation from the true NN is unlikely). Note that we do not attempt to discover the global topology, which can be computationally demanding and practically unnecessary. Using GPS-based driving as an analogy, we never need to access the world map (global topology) to reach our destination; knowing where to make the next turn (local geometry as design bias) is sufficient for the goal of navigation. Similarly, manifold learning aims not to reconstruct the entire manifold but to make a good prediction about the new data $x_{new}$ from the local neighborhood.

## 4 IMPLICIT REPRESENTATION VIA RECURSIVE KERNEL TRANSFORMATION

Design bias via context dependency alone cannot solve the learning problem; it has to work together with an ideal representation of both content and context. From a CoM perspective, the delta measure offers such an optimal (in the sense of robustness) representation that the probability measure concentrates on a single location. For sensory systems, Dirac's delta function often represents an ideal response of an organism's matched filter (MF) to the uncertain environment Wehner (1987). Using MF as the kernel in the latent space, we show in this section that kernel transformation can lead to the local concentration of probabilistic measure. Similar to the Gibbs measure in thermodynamic systems, the delta measure in neurodynamic systems can convert any explicitly formulated optimization problem into an implicitly defined representation in the latent space with over-parameterization. Recursive application of kernel transformation can serve as a non-Markovian model for context encoding of the working/episodic memory Baddeley (1992).

**Implicit Representations via Delta Measures**: The basic idea underlying implicit representation is that SLAM model (Fig. 2b) can implement a cortical transformation (change of coordinates), which extends Ballard's idea of generalized Hough transform (GHT) Ballard (1981) for detecting arbitrary shapes. GHT is a class of generalized matched filters (MFs) whose ideal response approximates Dirac's delta function Sanders & Yokoyama (2012). We note that the accumulation of voting results by GHT asymptotically approaches the delta measure or the path integration result in the latent space (parameterized by $\Theta$):

$$R_f(\Theta) = \int_{x \in R^n} f(x)\delta(S(x; \Theta))dx, \tag{1}$$

where $S_{\Theta}(x)$ denotes the kernel (e.g., implicit shape function Leibe et al. (2008)) and $f(x)$ is the function representing the sensory observation (stimulus). An important benefit of implicit representations via delta measures is that they can easily incorporate design bias or contextual variables into the local CoM in the latent space. If we use variable $\Phi$ to denote the set of contextual variables, the path integral transform can be rewritten into:

$$R_f(\Theta, \Phi) = \int_{x \in R^n} f(x)\delta(S_f(x; \Theta, \Phi))dx, \tag{2}$$

where $S_f$ is the generalized kernel, and we note the peer relationship of parameters in the latent space between content ($\Theta$) and context ($\Phi$).

Separating context from content in representation is conceptually similar to where-and-what separation in time-frequency analysis. The fundamental uncertainty principle dictates that one cannot simultaneously localize an event in paired measurements (e.g., position and momentum, space and frequency). We believe that kernel transformation in Eq. (2) obeys a similar limit - i.e., the CoM cannot be simultaneously localized in $\Theta$ and $\Phi$. Unlike explicit representations such as Markov Random Field (MRF) and the corresponding Gibbs measure, implicit representations via delta measures obey different organizational principles. MRF-based prior is a graphical model focusing on spatially local interaction among different nodes; the Gibbs potential (a.k.a., partition function) is simply a global summation of local functions. In complex systems, more is different Anderson (1972) - neurodynamic systems are characterized by hysteresis or memory-related path dependency. Macroscopic states of neurodynamic systems such as canonical ensembles have emergent properties, which are more than the sum of spatially local interactions.

Our understanding of emergent properties is still limited. Nonlinearity and predictability of complex systems are often at odds. How can the brain state be completely deterministic in its default mode (i.e., sleep) but complex yet still predictable in its perturbation mode (i.e., wake) Buzsáki (2006). Our reasoning here is based on two new insights: 1) based on the CoM theory Talagrand (1995), a set of measure $\geq \frac{1}{2}$ will ensure its $\epsilon$-ball expansion covers almost the entire space. To satisfy

this condition, delta measures must work together with design bias to characterize the macroscopic states of mammalian brains. 2) SLAM is known for its capability of establishing the correspondence between where (positions) and what (landmarks). The conceptual similarity between SLAM (updating and prediction) and GHT (voting and accumulation) inspired us to reinterpret SLAM as an agent specialized in kernel transformation that glues content (what) with context (where) into deictic codes Ballard et al. (1997). Connecting these two lines of reasoning, we conjecture that memory-based complex systems solve the prediction problem by localizing CoM in bilateral representations such as content and context. Formally, we have the following result.

**Proposition 2. (Bilateral implicit representation exploiting the CoM).**

*Any explicitly formulated approximation of $f(\boldsymbol{x}; \boldsymbol{\Theta})$ along with contextual variable $\boldsymbol{\Phi}$ admits a bilateral implicit representation by a kernel transformation $R_f(\boldsymbol{\Theta}, \boldsymbol{\Phi}) = \int_{\boldsymbol{x} \in R^n} f(\boldsymbol{x}) R_f(\boldsymbol{\Theta}, \boldsymbol{\Phi}) d\boldsymbol{x}$ that pushes the probabilistic measure to concentrate locally in the latent space. To exploit the CoM, the probability for a conditional event $\boldsymbol{\Phi}|\boldsymbol{\Theta}$ needs to be above chance (i.e., $P(\boldsymbol{\Phi}|\boldsymbol{\Theta}) \geq \frac{1}{2}$).*

The above proposition offers a new framework for the characterization of macroscopic states in complex systems - i.e., based on the mixture of Gaussian as in Algorithm 1. Conditioned on the location in the latent space parameterized by $\boldsymbol{\Theta}$, the probabilistic measure will be above the chance (so winner takes all). The key new observation is that both content and context of memory in neurodynamic systems can be implicitly represented by delta measures in the latent space. In mammalian brains, time-locking based PNGs represent a new class of canonical ensembles characterized by hysteresis in the phase/latent space (i.e., the broken symmetry of synaptic connections or causality principle - forward associations are stronger than backward ones Buzsáki (2006)). The complexity arises from nonlinear interaction among PNGs; while predictability is guaranteed by the reproducibility of macroscopic states (when the above-chance threshold is reached). Neurodynamic systems exploit the blessing of dimensionality to encode both content and contextual information into PNG-based macroscopic states with good generalization properties thanks to the counter-intuitive CoM phenomenon.

Simultaneous encoding of content and contextual information by the delta measures can reduce the overall complexity of the solution space for SbG learning, effectively localizing the search to high-probability regions (e.g., to facilitate implicit sampling Morzfeld (2015)). Another computational benefit of implicit representations is the straightforward calculation of marginal distributions via aliasing operators McCallum (1995) or more generally integral transforms. Unlike sequential importance sampling Liu, delta measures can readily separate content from context variables by integral transforms as following.

$$R_f(\boldsymbol{\Theta}) = \int_{\boldsymbol{\Phi}} R_f(\boldsymbol{\Theta}, \boldsymbol{\Phi}) d\boldsymbol{\Phi}, R_f(\boldsymbol{\Phi}) = \int_{\boldsymbol{\Theta}} R_f(\boldsymbol{\Theta}, \boldsymbol{\Phi}) d\boldsymbol{\Theta} \quad (3)$$

The above integral transforms could represent a computational abstraction of memory consolidation procedure during sleep when the hippocampus generalizes episodic memory into semantic memory and shifts it back to the neocortex Buzsáki & Moser (2013).

**Hierarchical Context Encoding via Recursive Kernel Transformation.** Complex systems are known for "carrying their history on their backs." Prigogine & Stengers (2018). Markov processes, as characterized by their memoryless properties, often fail to characterize the phenomenon of hysteresis in biological systems. Following geometric intuition in the puzzle and less-is-more heuristics, we have constructed a new non-Markovian sequence model for encoding episodic memory, as shown in Fig. 4. The key idea behind our construction lies in the following observations: 1) observed states $\mathbf{X}$ encode both sensory and motor information as content ($\boldsymbol{\Phi}$) and context ($\boldsymbol{\Theta}$) variables; 2) unobserved states recursively encode the memory (content and context) information into $\boldsymbol{z}$'s. Both $\mathbf{X}$'s and $\boldsymbol{z}$'s are macroscopic states of neurodynamic systems; unlike Gibbs measure in thermodynamic systems, we conjecture that *delta measure* is a more appropriate characterization of the probability density in the phase space. As soon as the probability of some event encoded by macroscopic state (e.g., PNGs Izhikevich (2006)) reaches above the threshold of $\frac{1}{2}$, the CoM phenomenon ensures its reproducibility in high dimensions or good generalization property for learning.

As advocated in Baldassano et al. (2017), memory formation is a hierarchical process covering multiple time scales. The recursive kernel transformation introduced here is only the first step toward modeling this complex system of memory formation - the first 100ms of sensory perception. Such

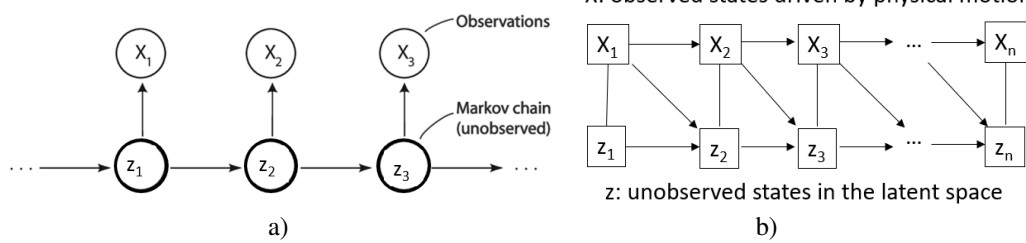

Figure 4: Sequential modeling of working memory: a) hidden Markov chain; b) non-Markov memory model recursively encodes contextual variable $X$ induced by motion into the hidden states $z$.

recursive kernel transformation offers a biologically plausible implementation of the parts-to-whole model Hinton (2023) by delta measures. At each level $z_k$, cortical columns, as abstracted by SLAM, accumulate the voting (MFs) results into a delta function, which serves as the landmark/what pointer to the next level $z_{k+1}$ (a newly defined latent space). Note that contextual information $x_k$ (parameterized by $\Theta$) is simultaneously encoded along with $z_k$ (parameterized by $(\Theta, \Phi)$). Such map-in-map hierarchy mirrors the hierarchical organization of the physical world Hawkins (2021).

The key insight underlying recursive kernel transformation is that fractal-like geometry can be a conceptual framework for hierarchical context encoding. Since features (e.g., shape, color, and texture) are implicitly coded into the kernel of varying dimensions, the delta measure (the ideal response of MF) becomes the bridge connecting feature representations across different scales. More specifically, we can generalize Eq. (1) by introducing the scale parameter $s$ ($s = 1, 2, ..., n$ corresponds to the subscripts in Fig. 4b):

$$R_f^{(s+1)}(\Theta, \Phi) = \int_{\boldsymbol{x} \in R^n} f(\boldsymbol{x}) R_f^{(s)}(\Theta, \Phi) d\boldsymbol{x}, \tag{4}$$

where recursion is defined with respect to the kernel from a local (e.g., simple edges) to a global (holistic concept) scale. When connected with CoM Talagrand (1995), we note that recursive kernel transformation in Eq. (4) conceptually extends the diectic codes Ballard et al. (1997) into a fractal-based pointer framework - i.e., the position of a fine-scale object in the coarse-scale map is the pointer or contextual variable. The sequential modeling of the working memory model in Fig. 4 can be interpreted as traversing along the multi-scale map-in-map hierarchy, so the conditional probability (after multi-scale conditioning) can reach above the chance ($\frac{1}{2}$). To conclude this section, we summarize our findings into the following proposition.

**Proposition 3. (Map-in-Map Hierarchy via Multiscale Context)**.

*To mirror the nested structure in the physical world, landmarks in a coarse-scale map can be implicitly represented by the voting and accumulation of landmarks in a fine-scale map. Such fractal-like contextual dependency across different scales ensures that the conditional probability $P(R_f^n | R_f^{n-1}, ..., R_f^1) \geq \frac{1}{2}$.*

*Sketch of the proof.* Using the causality principle, we can show that $P(R_f^n | R_f^{n-1}, ..., R_f^1) > P(R_f^{n-1} | R_f^{n-2}, ..., R_f^1) > ... > P(R_f^2 | R_f^1) > 0$. That is, multi-scale conditioning gradually pushes the CoM toward passing the chance threshold.

## 5 BIOLOGICAL CONNECTIONS

In this section, we make connections between implicit representations and existing knowledge in neuroscience. Our perspectives will be based on the well-accepted hypothesis that sensory systems of animals are adapted to the external environment through both evolutionary and developmental processes Simoncelli & Olshausen (2001).

**Phylogenetic continuity of navigation and memory.** The discovery of place cells O'keefe & Nadel (1978) marked an important milestone in our understanding of the brain mechanism. A clear

relationship between overt behavior (navigation) and high-level associative structure (gnostic units) was established for the first time. The construction of cognitive maps Whittington et al. (2022) from dead reckoning by the hippocampus through motor actions offers a convincing argument for the importance of temporal context in symmetry breaking. That is, to learn the concept of space (via spatial maps), an organism has to collect asymmetric episodic memory that is context dependent first. Then context-dependent memory (e.g., unidirectional place cells) is consolidated into context-independent memory (e.g., omnidirectional place cells) what has been known as semantic memory. Such restoration of plane symmetry is a universal strategy of generalization discovered by nature. With the memory of landmarks (stored by place cells), an organism can solve the navigation problem more efficiently than dead reckoning. It has been shown in Muller et al. (1996) that the minimization of the total synaptic resistances along a path in the latent (neural) space solves the shorted path problem assuming a densely connected network.

**Object Recognition by Ventral Stream.** An interesting analogy exists between spatial navigation and object recognition (the identification task) because they can be connected by changing coordinates from egocentric to allocentric Pouget & Sejnowski (1997). The saccadic movement of eyes for object recognition is conceptually similar to the physical movement of an organism in spatial navigation. Both behaviors are responsible for the generation of episodic memory that simultaneously encodes context and content information. However, the role of what and where is swapped from spatial navigation to object recognition. In spatial navigation, location or where is the task objective (content), and navigation direction is the context, while in object recognition, edge orientation is associated with content representation because location or where becomes the contextual variable. Just like the generalization achieved by memory consolidation (from unidirectional to omnidirectional place cells), a similar mechanism (previously known as the aliasing operator McCallum (1995)) can be responsible for context-invariant object recognition along the ventral stream. The increased complexity arises from the modeling of cortical transformation Keller & Mrsic-Flogel (2018) by recursive kernel transformations. Such map-in-map hierarchy is consistent with the local subspace untangling hypothesis in DiCarlo et al. (2012), which gradually untangles the object manifold along the ventral stream ($V1 \rightarrow V2 \rightarrow V4 \rightarrow IT$).

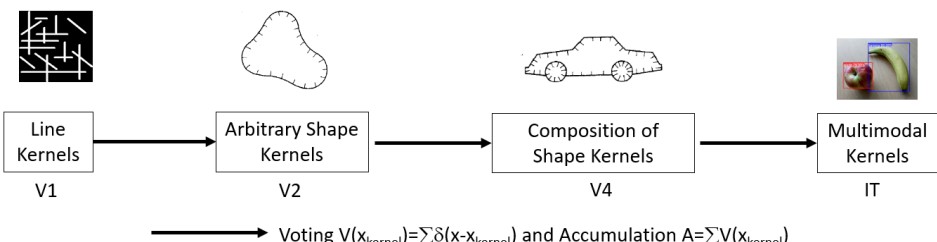

Figure 5: Computational modeling of the ventral stream for object recognition (100ms) via recursive kernel transformation. Note that each stage only involves two canonical operations: voting (linear MF) and accumulation (nonlinear thresholding generates the delta output, which becomes the input to the next level).

## 6 CONCLUSIONS

Learning is easy; representation is hard. The evolution of mammalian brains documents valuable hints about the mechanism of natural intelligence. Navigation-memory analogy inspires us to use SLAM agent for computationally modeling cortical columns. Hippocampal-cortical coupling implements an efficient indexing system to support the conversion between episodic and semantic memory. Implicit representation based on design bias and kernel transformation marks the first step toward unlocking the secret of memory encoding and consolidation. CoM phenomenon, despite being counter-intuitive, might be the Rosetta stone for discovering a universal cortical processing algorithm as predicted by V. Mountcastle. Location, location, location - a multi-scale extension of deictic codes pointing content and context bilaterally and hierarchically in the latent space - might be the secret for representation learning in nature.

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
