# OpenReview forum: "Location, Location, Location: Design Bias with Kernel Transformation"
_ICLR.cc/2025/Conference — ICLR 2025 Conference Withdrawn Submission_

### Official Review · Reviewer_MhEk · 2024-10-31

**Soundness:** 1
**Presentation:** 1
**Contribution:** 1
**Rating:** 1
**Confidence:** 3

**Summary:**

The authors seem to aim to develop a new manifold learning model based on inspiration from hippocampal-cortical computation for navigation.  Some formal developments are provided with biological connections.  No experiments have been done.

**Strengths:**

As the manuscript has a severe unclarity, I could not find any strength on either originality or significance.

**Weaknesses:**

The manuscript has an overall low quality of presentation and lacks coherent logic throughout the document.  Most claims about originality, soundness, and performance are obfuscated with backgrounds and citations whose relevances are unclear.  Many different materials are presented but these are poorly organized so that how these are related to each other is not clear.  Overall, the manuscript does not seem to contain prominently interesting claims with sufficiently convincing supports.

Reading the first half of the paper, I interpret (though not clear) that the main purpose is to propose a manifold learning based on biological navigation in hippocampus.  Indeed, Section 3 gives a manifold learning algorithm.  However, as far as I can see, this seems the most obvious greedy algorithm that tries to just connect between nearest points to form a manifold, which seems to have no technical interest at all.  Otherwise, it may be just an algorithmic outline without sufficient concreteness, for which assessing novelty is not possible.   In any case, the authors did not implement their algorithm or empirically test any possible advantage of their idea.  Therefore, even if the algorithm has any novelty, the significance is unclear.  They also argue about design bias, but the overall arguments are obscure and I could not find any conceptually novelty with respect to classical Bayesian approaches.

Some discussions on biological connection are given especially in Sections 1 and 5.  However, these seem to be a list of general neuroscience knowledges (hippocampal navigation system and visual ventral stream) without much new insight.  I do not see how these lead to better understanding of evolution of brain as they claim.  In other parts, the authors mention relevance to sleep/wave as well as working memory, though the connection is unconvincing.

Some formal developments on recursive kernel transformation are provided in Section 4.  This part is a bit technical and I could not follow well.  At least, I could not see at all how exactly those are connected to the materials discussed in previous sections like the manifold learning.  Two propositions are given, but at least one lacks proof (they have no supplementary materials).

**Questions:**

None.

---

### Official Review · Reviewer_wUk4 · 2024-11-02

**Soundness:** 1
**Presentation:** 1
**Contribution:** 1
**Rating:** 1
**Confidence:** 3

**Summary:**

The work presented here is incomprehensible - The paper contains a string of concepts and ideas that are not meaningfully connected. Most sentences contain odd formulations and there is simply no discernible insight that can be taken from this paper.

**Strengths:**

The work is fundamentally incomprehensible, with no discernible strenghts.

**Weaknesses:**

The authors string together a wealth of unconnected ideas with wholly inadequate and confusing explanations. This work is aiming to find a connection between, on one hand, interactions of the mammalian hippocampus and neocortical interactions, and on the other hand techniques from various ML fields.
The writing is opaque, confusing, and not unlike the hallucinations of a LLM.

A few examples are:
"does there exist a universal and optimal solution to transform folded data manifolds as shown in Fig. 2 into linearly separable Dirac’s delta functions in the latent space? A moment of thought shows that the answer is Yes but nontrivial. For example, the kernel trick or neural network will not work here because there is no generic rule for kernel construction or neural network that works for data manifolds with arbitrary topology and varying folding strategies (note that a different way of paper folding will change the definition of geodesic distance in the folded space)"

"Our understanding of emergent properties is still limited. Nonlinearity and predictability of complex systems are often at odds. How can the brain state be completely deterministic in its default mode (i.e., sleep) but complex yet still predictable in its perturbation mode (i.e., wake) Buzsaki (2006). Our reasoning here is based on two new insights: 1) based on the CoM theory Talagrand (1995), a set of measure ≥ x will ensure its ε-ball expansion covers almost the entire space."

"Figure 2: Toy examples of data manifolds in 2D - after folding into 3D (Image credit: adapted from the Internet), does there exist a universal solution to represent and separate different classes (data points labeled by different colors) regardless of the folding strategy? In this paper, we provide affirmative answers and show its connections with universal learning via implicit representations."

"CoM phenomenon, despite being counter-intuitive, might be the Rosetta stone for discovering a universal cortical processing
algorithm as predicted by V. Mountcastle. Location, location, location - a multi-scale extension of deictic codes pointing content and context bilaterally and hierarchically in the latent space - might be the secret for representation learning in nature"

The sentences seem superficially meaningful, but as a whole there is little else than randomly strung together bits of text.
There are also frequent formatting issues, the notation and pseudocode (Algorithm 1) has trivial errors and inconsistent symbols. To be able to review and judge this work, a major revision is necessary to help the reader interpret the presented ideas.

**Questions:**

See above - a major revision is necessary in order to ask meaningful questions about this work.

---

### Official Review · Reviewer_BFaK · 2024-11-04

**Soundness:** 1
**Presentation:** 1
**Contribution:** 1
**Rating:** 1
**Confidence:** 4

**Summary:**

The paper proposes a framework for understanding manifold learning in the brain through local linearization, where cortical columns act as SLAM agents to learn local maps. Extending Ballard’s generalized Hough transform (GHT), the authors present this concept within a novel SbG (Specification-before-Generalization) learning model. They hypothesize that concentration of measure (CoM)-based bias can solve the manifold learning problem.

**Strengths:**

The authors present a potentially intriguing idea by linking the brain’s navigation and memory processes to a mathematical framework for manifold learning. This conceptual framework draws connections between brain structures (e.g., cortical columns) and computational models, particularly the SLAM framework. While speculative, some analogies presented may serve as inspiration for further work.

**Weaknesses:**

The paper’s main contributions are primarily speculative, with limited empirical validation or rigorous technical proofs to substantiate the claims. This reliance on analogy and conjecture significantly limits the paper's scientific value. Algorithm 1, appears to be the main technical contribution, so do not just sketch Algorithm 1 but implement it! Demonstrate that it works “to achieve optimal classification asymptotically” and compare to other classification algorithms. For example, you reference k-nearest neighbors (kNN) in Algorithm 1 so compare your algorithm to other variations of kNN (Halder et al. 2024 “Enhancing K-nearest neighbor algorithm: a comprehensive review and performance analysis of modifications”). There are currently no quantitative comparisons to any other classification algorithms, which makes it hard to appreciate your contributions.

* Key propositions lack formal proofs, and certain statements—such as the conjecture that “memory-based complex systems solve the prediction problem by localizing CoM in bilateral representations such as content and context”—appear overly broad and scientifically implausible. It is hard to see how a single mechanism could reliably address the prediction problem across any memory-based complex system. To improve your paper, consider providing specific examples or scenarios where your proposed mechanism would apply, and clarify the scope of "memory-based complex systems" you are referring to.

* Several issues of clarity also detract from the work. The use of unexplained technical jargon and dense conceptual terms makes it difficult to parse the core claims and understand the proposed contributions. Definitions and technical explanations of foundational terms are sparse, and the writing lacks an organizational structure that would aid comprehension. Including a glossary or appendix to clarify terms (e.g. generalized Hough transform (GHT), concentration of measure (CoM), matched filter (MF), specification-before-generalization (SbG)) and importantly, how they contribute to the larger goals of the paper, would improve readability.

**Questions:**

* Could you clarify how your approach deviates from or improves upon nearest-neighbor methods?

* Why are there no results demonstrating that your proposed method is capable of learning practical tasks?

* What are the technical and empirical bases for your claims about delta measures and CoM, especially regarding their relevance to brain modeling?

* To support accessibility, consider reorganizing the paper for clarity, defining terminology early on, and providing foundational explanations for technical concepts.

---

### Official Review · Reviewer_iLPR · 2024-11-04

**Soundness:** 1
**Presentation:** 1
**Contribution:** 1
**Rating:** 3
**Confidence:** 1

**Summary:**

Unfortunately, I do not think I can write an appropriate review for this paper, as it was difficult for me to understand. I would be unable to write a summary for it that I am confident in.

Several of the neuroscience papers referenced are cited in strange places (and contexts). I feel as though they are incorrectly cited, but because it is hard to know how they motivated the quantitative theory, I cannot be certain. Some of the other papers, e.g., Dissanayke et al., 2001, are beyond my expertise. There are also some papers cited, e.g., Sanders & Yokoyama, 2012 on "Reverse Mathematics", that I have never heard of before.

Because I could not understand this paper, I cannot recommend its acceptance. However, note that I have placed a '1' on the confidence scale, and have mentioned this to the AC.

**Strengths:**

See summary.

**Weaknesses:**

See summary.

**Questions:**

See summary.

---

### Note · Authors · 2024-11-12

I have read and agree with the venue's withdrawal policy on behalf of myself and my co-authors.